# Resting behaviour of malaria vectors in highland and lowland sites of western Kenya: Implication on malaria vector control measures

Maxwell G. Machani[1,2], Eric Ochomo[1], Fred Amimo[2], Jackline Kosgei[1], Stephen Munga[3], Guofa Zhou[4], Andrew K. Githeko[3], Guiyun Yan[4], Yaw A. Afrane[5]*

1 Entomology Section, Centre for Global Health Research, Kenya Medical Research Institute, Kisumu, Kenya, 2 School of Health Sciences, Jaramogi Oginga Odinga University of Science and Technology, Bondo, Kenya, 3 Centre for Global Health Research, Kenya Medical Research Institute, Kisumu, Kenya, 4 Program in Public Health, College of Health Sciences, University of California, Irvine, California, United States of America, 5 Department of Medical Microbiology, University of Ghana Medical School, University of Ghana, Accra, Ghana

* yaw_afrane@yahoo.com

**Data Availability Statement:** All relevant data are within the paper and its two Supporting Information files.

## Abstract

### Background

Understanding the interactions between increased insecticide resistance and resting behaviour patterns of malaria mosquitoes is important for planning of adequate vector control. This study was designed to investigate the resting behavior, host preference and rates of *Plasmodium falciparum* infection in relation to insecticide resistance of malaria vectors in different ecologies of western Kenya.

### Methods

*Anopheles* mosquito collections were carried out during the dry and rainy seasons in Kisian (lowland site) and Bungoma (highland site), both in western Kenya using pyrethrum spray catches (PSC), mechanical aspiration (Prokopack) for indoor collections, clay pots, pit shelter and Prokopack for outdoor collections. WHO tube bioassay was used to determine levels of phenotypic resistance of indoor and outdoor collected mosquitoes to deltamethrin. PCR-based molecular diagnostics were used for mosquito speciation, genotype for knockdown resistance mutations (1014S and 1014F) and to determine specific host blood meal origins. Enzyme-linked Immunosorbent Assay (ELISA) was used to determine mosquito sporozoite infections.

### Results

*Anopheles gambiae* s.l. was the most predominant species (75%, n = 2706) followed by *An. funestus* s.l. (25%, n = 860). *An. gambiae* s.s hereafter (*An. gambiae*) accounted for 91% (95% CI: 89–93) and *An. arabiensis* 8% (95% CI: 6–9) in Bungoma, while in Kisian, *An. arabiensis* composition was 60% (95% CI: 55–66) and *An. gambiae* 39% (95% CI: 34–44). The

**Funding:** This study was supported by grants from the National Institute of Health (R01 A1123074, U19 AI129326, R01 AI050243, D43 TW001505). There was no additional external funding received for this study. The funders had no role in study design, data collection and analysis, decision to publish, or preparation of the manuscript.

**Competing interests:** The authors have declared that no competing interests exist.

**Abbreviations: BBI**, bovine blood index; **CSP**, circumsporozoite protein; **EIR**, entomological inoculation rate; **ELISA**, enzyme-linked immunosorbent assay; **F1**, first generation offspring; **HBI**, human blood index; **IRS**, indoor residual spray; **KDR**, Knockdown resistant gene; **LLINs**, long-lasting insecticidal nets; **PCR**, polymerase chain reaction; **PSC**, pyrethrum spray catch; **Vgsc**, voltage-gated sodium channel.

resting densities of *An. gambiae* s.l and *An. funestus* were higher indoors than outdoor in both sites (*An. gambiae* s.l; $F_{1, 655} = 41.928$, p < 0.0001, *An. funestus*; $F_{1, 655} = 36.555$, p < 0.0001). The mortality rate for indoor and outdoor resting *An. gambiae* s.l F1 progeny was 37% (95% CI: 34–39) vs 67% (95% CI: 62–69) respectively in Bungoma. In Kisian, the mortality rate was 67% (95% CI: 61–73) vs 76% (95% CI: 71–80) respectively. The mortality rate for F1 progeny of *An. funestus* resting indoors in Bungoma was 32% (95% CI: 28–35). The 1014S mutation was only detected in indoor resitng *An. arabiensis*. Similarly, the 1014F mutation was present only in indoor resting *An. gambiae*. The sporozoite rates were highest in *An. funestus* followed by *An. gambiae*, and *An. arabiensis* resting indoors at 11% (34/311), 8% (47/618) and 4% (1/27) respectively in Bungoma. Overall, in Bungoma, the sporozoite rate for indoor resting mosquitoes was 9% (82/956) and 4% (8/190) for outdoors. In Kisian, the sporozoite rate was 1% (1/112) for indoor resting *An. gambiae*. None of the outdoor collected mosquitoes in Kisian tested positive for sporozoite infections (n = 73).

## Conclusion

The study reports high indoor resting densities of *An. gambiae* and *An. funestus*, insecticide resistance, and persistence of malaria transmission indoors regardless of the use of long-lasting insecticidal nets (LLINs). These findings underline the difficulties of controlling malaria vectors resting and biting indoors using the current interventions. Supplemental vector control tools and implementation of sustainable insecticide resistance management strategies are needed in western Kenya.

## Background

Malaria still remains a major public health concern in sub-Saharan Africa, responsible for an estimated 228 million cases and 405,000 deaths despite the massive investments in scaling-up indoor anti-vector interventions [1]. Remarkable advances in the fight against malaria have been achieved within the past decade mainly through the massive scale-up of long-lasting insecticide-treated nets (LLINs) and indoor residual spraying (IRS) in many localities [2,3]. Despite the increased efforts, it is worrying that no significant progress has been made in reducing global malaria cases in the year 2015–2017 period [4], with some regions in sub-Saharan Africa, previously reported to experience a resurgence of malaria including western Kenya [5]. This transmission recurrence is partly attributed to the emergence of insecticide resistance and behavioural modification that have arisen as an adaptation by mosquitoes in response to high use of insecticides for vector control [6,7]. All these factors have the potential to weaken malaria control programs thus posing a serious threat in the fight against malaria [8].

The current vector control interventions take advantage of susceptible mosquito behaviors. These interventions are based on the observation that malaria vectors prefer to bite humans indoors late at night and often rest inside houses after blood feeding hence, they will be exposed to sufficient levels of insecticides which will either kill them or reduce their longevity thus affecting their vectorial capacity [4]. In sub-Saharan Africa insecticide-treated net (ITN) ownership is estimated to have increased from 3% in 2000 to 83% in the period of 2015–2017 [4]. In Kenya, the government rolled out the universal bed net programme where every two persons in a household were provided with a free ITN. The ITN ownership rose from 12.8% in 2004 to over 80% in 2015[9,10] The increased use of indoor interventions may pose stress on

the indoor feeding and resting of malaria vectors leading to either behavioural defense [11] or physiological defense [6]. Malaria vectors have been shown to adapt to changing environment due to either behavioural avoidance or selection of mutations and recombination that favour their survival in the presence of insecticides threatening the efficacy of the current indoor-based vector control tools [6,12] and the resulting increase in residual transmission [13]. Insecticide resistance is common in sub-Saharan Africa with some regions reporting resistance to all classes of insecticides [14,15]. In Kenya, the target site and metabolic resistance mechanisms play a major role in pyrethroid resistance [16,17]. The primary malaria vectors in Kenya belong to *An. gambiae* complex and *An. funestus* group due to their anthropophilic and endophilic behaviours that makes them be more efficient in malaria transmission [18–20]. With the scale-up of indoor-based vector control tools mosquitoes have changed behaviours; some are biting and resting indoors whilst others have changed to prefer biting and resting outdoors. Behavioral modifications including changes in biting time and location [21–23], changes in host choice and shift from endophilic (i.e. resting in houses) to exophilic (i.e. resting outdoors) behavior have been associated with long-term use of insecticide-based interventions [24,25]. Knowledge of the resting habits of resistant vectors and their feeding preference may predict the intensity of malaria transmission. It is hypothesized that insecticide-resistant malaria vectors could bite and rest indoors in the presence of interventions whilst susceptible ones bite and rest outdoors. Additionally, behaviors of malaria vectors have been shown to differ on small geographical scales, further complicating malaria elimination efforts[26]. Understanding how the resting habits of malaria vectors change in response to current indoor-based vector control interventions is important for sustaining vector control. These behavioral modifications and physiological resistance in most of the malaria vectors have been shown to contribute to maintaining malaria transmission [12,27].

In order to improve vector control intervention strategies, it is crucial to characterize the behavioral patterns of each species of a particular vectorial system in their specific settings over time and in a range of environmental changes, especially with increasing pyrethroid resistance. The objective of this study was to investigate the species diversity of malaria vectors, their resting behavior, and the distribution of infections in two ecological settings of western Kenya with different levels of insecticide resistance. This information could provide a better understanding of the interactions between increased insecticide resistance phenotypes in field malaria vector population and the subsequent resting behaviour patterns in the presence of the current indoor intervention.

## Methods

### Study site

The study was carried out in Kisian (00.02464˚S, 033.60187˚E, altitude 1,280–1,330 m above sea level), Kisumu county and Bungoma (00.54057˚N, 034.56410˚E, altitude 1386–1,545 m above sea level) in Bungoma County, all in western Kenya. Kisian is located in the lowland area around Lake Victoria in western Kenya. A*n. gambiae sensu stricto* (s.s.*), An. arabiensis* and *An. funestus* are the main vectors of malaria in this region [17,28]. Bungoma County is located in malaria epidemic-prone highland area. The sites experience a bimodal pattern of rainfall, with the long rainy season (April—July) which triggers peak malaria transmission period and the short rainy season (October-November) and year-to-year variation. The hot and dry season is from January to March which marks the lowest transmission period[5] Both sites are endemic for *Plasmodium falciparum* malaria. The malaria vector population in both sites include *An. gambiae* and *An. arabiensis* and *Anopheles funestus* [16,17]

## Mosquito sampling

Mosquitoes were sampled during the middle of the long dry season (February-March) and four weeks after the start of the long rainy season (May-July) in 2018. Indoor resting malaria vectors were sampled using pyrethrum spray catches (PSCs) in sixty (60) randomly selected houses from 06:00 to 09:00 h [23]. The prokopack aspirator (John W Hock, Gainesville, FL, USA) was used to collect mosquitoes resting indoors and outdoors from the selected houses every morning. For indoor collections, mosquitoes resting on the walls and under the roof of the house or ceiling, under the beds were systematically aspirated. Outdoor sampling points included granaries, kitchens and evening outdoor human resting points. Outdoor resting mosquitoes were additionally collected from pit shelters constructed according to Muirhead-Thomson's method [29] within 20 m of each selected house, resting mosquitoes in the cavities created in the pit shelter were collected from 06:00 to 09:00 h by using Proko-pack. Clay-pots were used to collect outdoor resting mosquitoes. The pots were placed then left outdoors from 18:00 to 06:00h at about 5m from the window of selected houses [30]. Resting mosquitoes in the pots were collected in the morning from 06:00 to 09:00h by plac-ing a white mesh from a mosquito cage over the mouth and agitating the mosquitoes inside the pot, causing them to fly and move to the cage [30,31]. The pot was checked for the remaining mosquitoes and were collected using an aspirator to a well-labeled paper cup. *Anopheline* mosquitoes were sorted morphologically according to the identification keys described by [32]. Female mosquitoes were further classified according to their gonotrophic status. Mosquitoes from each collection method were stored in separately labeled vials and preserved by desiccation.

Some of the collected indoor and outdoor resting mosquitoes that were either blood-fed or gravid from the two sites were kept in paper cups covered with moistened cotton towels and transported to the insectary at Kenya medical research institute in Kisumu. Gravid *An. gambiae* s.l and *An. funestus* s.l females were provided with oviposition cups. Eggs laid were allowed to hatch in spring water in small trays and larvae reared on a mixture of tetramin (fish food) and brewer's yeast provided daily under controlled standard insec-tary conditions with a temperature range of 26± 2˚C and 70% to 80% relative humidity. Emerging adults were provided with a 10% sugar solution until ready to be used for bioassay tests.

## WHO resistance bioassays

To assess susceptibility or resistance of F1 progeny of mosquitoes caught from different loca-tions(indoor and outdoor) and study sites, emerging female adults aged 2–5 days were exposed to 0.05% deltamethrin following the standard WHO tube test protocol[33] for 1 h. The knockdown time (KDT) of females was reported every 10 min during the 60 min exposure period. After the 1 h exposure, surviving mosquitoes were transferred to recovery tubes and provided with 10% sucrose for 24 h holding period. Mosquitoes alive 24 h after the 60-min insecticide-exposure time were classified as resistant. Mortality was scored after the 24 h recovery period.

## *Anopheline* species discrimination

Sibling species of the *An. gambiae* and *An. funestus* complexes were distinguished using con-ventional PCR [34,35]. DNA was extracted from mosquito legs and wings using ethanol pre-cipitation method [36].

### Detection of sporozoite infectivity

The head and thorax of individual mosquitoes samples collected were used to detect the presence of *P. falciparum* sporozoites using enzyme-linked immunosorbent assays (ELISA) method as described by Wirtz et al. [37].

### Detection of blood meal sources using polymerase chain reaction (PCR)

The abdominal section of blood-fed *Anopheles* mosquitoes were cut transversely between the thorax and the abdomen. Genomic DNA was extracted from mosquito abdomens using ethanol precipitation method as described by Collins et al. [36]. One universal reverse primer and five animal-specific forward primers (human, cow, goat, pig, and dog) were used for amplification of cytochrome b gene, encoded in the mitochondrial genome to test for specific host blood meal origins using conventional PCR [38].

### Genotyping for kdr mutations

DNA was extracted from adult *An. gambiae* and *An. arabiensis* mosquitoes as earlier described [34]. Real-time (RT) PCR was used to detect mutations at amino acid position 1014 of the voltage-gated sodium channel (Vgsc) using a modification of the protocol by Bass et al. [39]. Samples were genotyped for both Vgsc-1014S and 1014F kdr alleles.

### Scientific and ethical clearance

The institutional review board of Kenya Medical Research Institute granted ethical review and approval (Ref: KEMRI/SERU/CGHR/085/3434). Prior to the commencement of data collection, a detailed explanation of the aims, study procedures, risks and benefits were provided to community leaders and participants of each study site. Informed consent was obtained from the household heads. Participation was voluntary and household heads were free to withdraw from the study in case of any inconvenience.

### Data analysis

Resting density of Anopheline mosquitoes was calculated as the number of female mosquitoes per trap/night for each trapping method. Analysis of variance (ANOVA) was used to compare malaria vector density between indoor and outdoor locations. Chi-square was used to test the difference in seasonal abundance and malaria vector species composition between resting locations (indoor and outdoor). Human blood index (HBI) was calculated as the proportion of blood-fed mosquito samples that had fed on humans to the total tested for blood meal origins [40]. Bovine, goat, dog, and pig blood indices were also calculated in a similar way. Mixed blood meals were included in the calculation of blood meal indices [41].

The sporozoite infection rate (IR) expressed as the proportion of mosquitoes positive for *Plasmodium* sporozoite was calculated by dividing the number of sporozoite positive mosquitoes by the total number of mosquitoes assayed. The frequency of the resistance allele was calculated using the Hardy-Weinberg equilibrium test for kdr genotypes. Data were analyzed using R software packages.

## Results

### Indoor and outdoor *Anopheline* mosquito composition

A total of 2,706 and 860 female *Anopheline* mosquitoes were collected from Bungoma (highland site) and Kisian (lowland site) respectively during the study period. *Anopheles gambiae* s.l

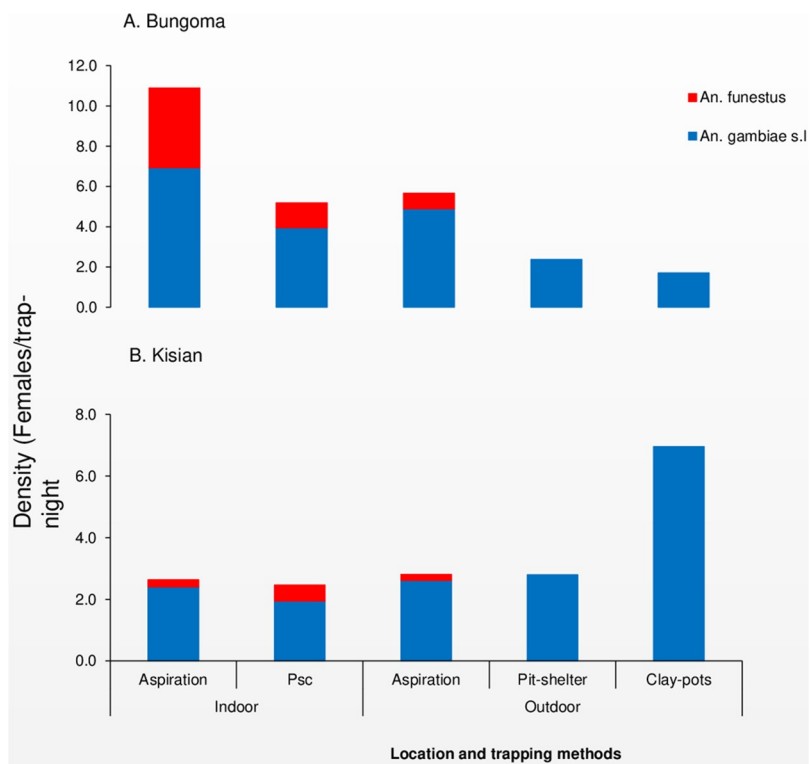

**Fig 1. Indoor and outdoor resting density of female *Anopheles* mosquitoes collected per trapping method A: Highland site (Bungoma) and B: Lowland site (Kisian), western Kenya.**

was the most abundant species accounting for 70% (95% CI: 68–71) in Bungoma and 91% (95% CI: 89–93) in Kisian followed by *An. funestus* 31% (95% CI: 29–32) and 10% (95% CI: 7–11) respectively. In Bungoma out of 1880 *An. gambiae* s.l collected, 85% (1606/1880) was caught resting indoors and 15% (274/1880) were caught resting outdoors. For *An. funestus*, 97% (798/826) were caught resting indoors and 3% (28/826) were caught outdoors. In Kisian, 58% (453/781) *An. gambiae* s.l were resting indoors and 42% (328/781) were caught resting outdoors. For *An. funestus*, 91% (72/79) were collected indoors and 9% (7/79) was caught resting outdoors (S1 Table). The mean indoor resting density of *An. gambiae* s.l from both sites was significantly higher than outdoor resting density ($F_{1, 655}$ = 41.928, p < 0.0001). The mean indoor resting density for *An. funestus* was also higher than outdoor resting density ($F_{1, 655}$ = 36.555, p < 0.0001) (Fig 1).

### *Anopheles gambiae* and *An. funestus* sibling species composition

For species identification, a sub-sample of 1,566 from both sites (1,172 *An. gambiae* s.l and 394 *An. funestus* s.l) were used to discriminate the sibling species. In Bungoma, 843 samples of *An. gambiae* s.l were analysed. Of these 91% (95% CI: 89–93) were *An. gambiae* and 8% (95% CI: 6–9) were *An. arabiensis*. Overall, of the three vector species, *An. gambiae* (66%, 95% CI: 64–70) was the predominant malaria vector in Bungoma followed by *An. funestus* (28%, 95% CI: 25–30) and *An. arabiensis* (5%, 95% CI: 4–7). There was a significant difference between indoor and outdoor locations in terms of mosquito species composition in Bungoma ($X^2$ = 122.96, *df* = 2, p < 0.0001). In Kisian, out of the 329 samples analysed 60% (95% CI: 55–66) were *An. arabiensis* and 39% (95% CI: 34–44) were *An. gambiae*. Overall *An. arabiensis* (49%,

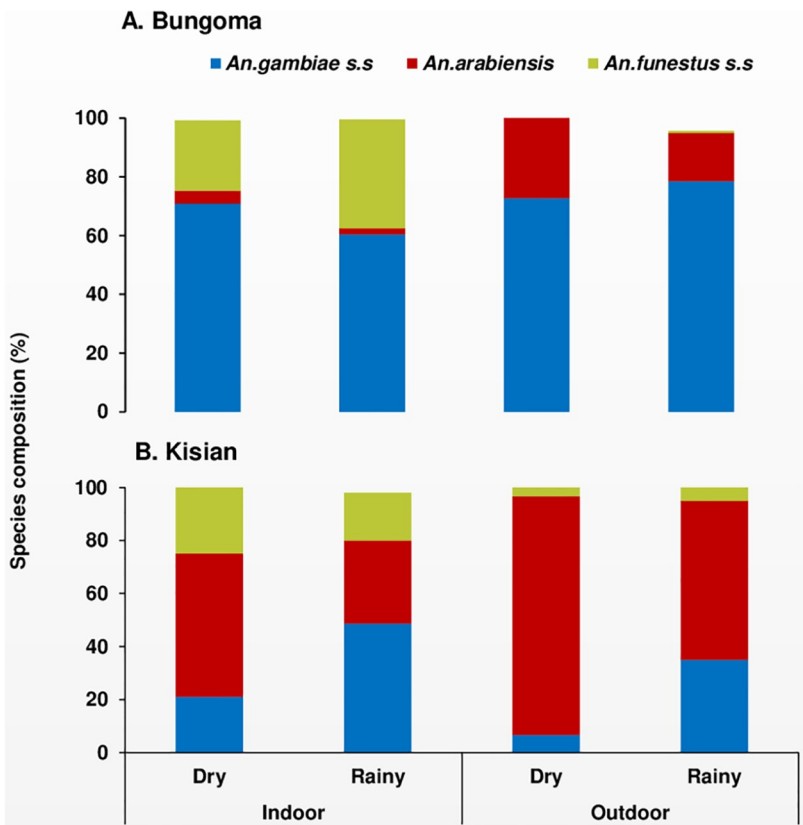

**Fig 2. Seasonal composition and *Anopheles* sibling species composition resting indoors and outdoors in A: Highland site (Bungoma) and B: Lowland site (Kisian), western Kenya.**

95% CI: 44–54) was the most abundant vector species followed by *An. gambiae* (32%, 95% CI: 27–36) and *An. funestus* (19%, 95% CI: 15–23, Fig 2). All the *An. funestus* s.l assayed from the two study sites were all *An. funestus* s.s (hereafter *An. funestus*).

In Bungoma, the seasonal composition of *An. gambiae and An. funestus* species was higher during the rainy season (57%, 95% CI: 53–61) and (67%, 95% CI: 63–73) compared to the dry season (43%, 95% CI: 39–47) and vs (32%, 95% CI: 25–37)($X^2 = 16.28$, *df* = 2, p < 0.0003) respectively. In contrast, in Kisian the overall seasonal prevalence of the three vector species composition was higher during the dry season than rainy season (*An. arabiensis*, 68% (95% CI: 61–76) and 32% (95% CI: 24–40), *An. funestus*, 63 (95% CI: 52–74) and 37% (95% CI: 21–42); $X^2 = 30.42$, *df* = 2, p < 0.0001, Fig 2).

## Phenotypic resistance

All the F1 mosquito populations tested from Bungoma and Kisian showed remarkable resistance to deltamethrin, with mortality rates ranging from 32–76% (Fig 3). Reduced mortality was observed for F1 progeny of *Anopheles gambiae* s.l resting indoors (37%, 95% CI: 34–39) than outdoors (67%, 95% CI: 62–69) in Bungoma. In Kisian the F1 progeny of *Anopheles gambiae* s.l resting indoors had lower mortality rates (67%, 95% CI: 61–73) than outdoors (76%, 95% CI: 71–80). Though the levels of deltamethrin resistance observed were higher for mosquitoes resting indoors compared to outdoors across the sites, there was no significant difference between the means ($F_{3, 28} = 1.391$, p < 0.266). The mortality rate for F1 progeny of *Anopheles funestus* resting indoors from Bungoma was 32% (95% CI: 28–35). Due to technical

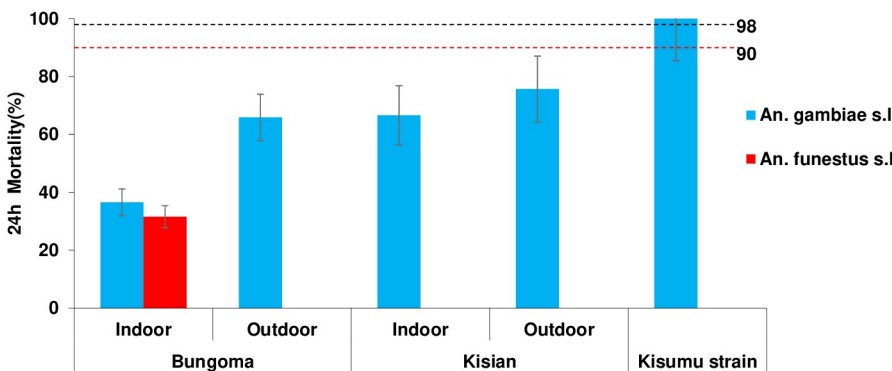

**Fig 3. Mortality rates of indoor and outdoor resting *Anopheles gambiae* s.l and *An. funestus* F1 progeny in standard WHO tube bioassays after exposure to 0.05% deltamethrin test papers and 24 hr recovery period.** Dotted lines represent upper (98%) and lower (90%) cut-offs for WHO classifications; values above the upper line indicate susceptibility and values below the lower red line indicate resistance (WHO, 2016). Error bars for the mean are shown.

difficulties in raising *An. funestus* and the small numbers collected resting outdoors, susceptibility test was not done in Kisian and for the outdoor population from Bungoma. The *An. gambiae s.s* Kisumu susceptible laboratory strain showed 100% mortality.

## Target site genotyping

In total 693 *Anopheles, gambiae* s.l samples were genotyped for the presence of Vgsc-1014S and 1014F mutations. In Bungoma, overall the frequency of Vgsc-1014S was 88% (290/328) and 6% (20/328) for Vgsc-1014F in *An. gambiae*, whereas only Vgsc-1014S was observed in *An. arabiensis* with a low frequency of 4% (2/52) (Table 1). The frequency of Vgsc-1014S and 1014F was 90 (177/195) vs 9% (20/195) respectively for indoor resting *An. gambiae*. Only Vgsc-1014S was observed for outdoor collections with a frequency of 85% (113/133). The Vgsc-1014S was the only kdr mutation observed in *An. arabiensis* resting indoors (10%, 2/20) and was not detected in the outdoor resting collections (S2 Table). In Kisian, the frequency of Vgsc-1014S in *An. gambiae* was 70% (89/125) and that of 1014F was 1% (2/125), whereas only Vgsc-1014S was observed in *An. arabiensis* with a frequency of 18% (36/188). The frequency of Vgsc-1014S mutation for *An. gambiae* resting indoors was 72% (78/107) and 61% (11/18) for outdoor collections. The same mutation was present in *An. arabiensis* collected from indoors 18% (20/110) and outdoors 18% (16/78). The Vgsc-1014F was only observed in *An. gambiae* caught resting indoors at a low frequency of 1%.

**Table 1. Frequency of *Kdr* mutations in *An. gambiae s.s* and *An. arabiensis* populations of western Kenya.**

| Site | Species | Location | *An. gambiae s.s* | | | *An. arabiensis* | | |
|---|---|---|---|---|---|---|---|---|
| | | | N tested | 1014S | 1014F | N tested | 1014S | 1014F |
| Bungoma | | Indoor | 195 | 90 | 9 | 20 | 10 | 0 |
| | | Outdoor | 133 | 85 | 0 | 32 | 0 | 0 |
| | *An. gambiae* | | 328 | 86 | 6 | | | |
| | *An. arabiensis* | | | | | 52 | 3.8 | 0 |
| Kisian | | Indoor | 107 | 72 | 1 | 110 | 18 | 0 |
| | | Outdoor | 18 | 61 | 0 | 78 | 18 | 0 |
| | *An. gambiae* | | 125 | 70 | 1 | | | |
| | *An. arabiensis* | | | | | 188 | 18 | 0 |

**Table 2. Blood meal origins of *Anopheline* mosquitoes collected from indoor and outdoor in Bungoma and Kisian, western Kenya.**

| Site | Blood-meal origins | An.gambiae s.s | | An. arabiensis | | An. funestus | |
|---|---|---|---|---|---|---|---|
| | | Indoor | Outdoor | Indoor | Outdoor | Indoor | Outdoor |
| Bungoma | **Number tested** | **422** | **39** | **24** | **8** | **111** | **4** |
| | Human | 229 (54) | 13 (33) | 6 (25) | 1 (13) | 72 (65) | 3 (75) |
| | Bovine | 60 (14) | 9 (23) | 16 (67) | 3 (38) | 14 (13) | 1 (25) |
| | Human+Bovine | 31 (7) | 4 (10) | 0 | 1 (13) | 9 (8) | 0 |
| | Other | 24 (6) | 4 (1) | 2 (8) | 1 (13) | 3 (3) | 0 |
| | Un-identified | 78 (19) | 9 (23) | 0 | 2 (25) | 13 (12) | 0 |
| | **HBI** | **65** | **46** | **25** | **25** | **75** | **75** |
| | BBI | 22 | 36 | 67 | 63 | 21 | 25 |
| Kisian | **Number tested** | **77** | **3** | **114** | **32** | **23** | **0** |
| | Human | 42 (56) | 2 (67) | 6 (5) | 0 | 18 (78) | 0 |
| | Bovine | 26 (34) | 0 | 104 (91) | 29 (91) | 4 (17) | 0 |
| | Human+Bovine | 3 (4) | 1 (33) | 2 (2) | 1 (3) | 1 (4) | 0 |
| | Other | 4 (5) | 0 | 1 (1) | 2 (6) | 1 (4) | 0 |
| | Un-identified | 2 (3) | 0 | 1 (1) | 0 | 0 | 0 |
| | **HBI** | **60** | **100** | **7** | **6** | **83** | **0** |
| | BBI | 40 | 33 | 93 | 97 | 22 | 0 |

Others: dog, goat, pig, or multi-source excluding human+bovine

HBI, human blood Index; BBI, bovine blood index

## Blood meal sources

A total of 857 blood-fed (719 *An. gambiae* s.l and 138 *An. funestus*) mosquito specimens were analysed for blood meal origins (Table 2). *An. funestus* was the most anthropophagic species and *An. arabiensis* the least from both sites. In Bungoma, the human blood index (HBI) for *An. gambiae*, *An. arabiensis* and *An. funestus* caught resting indoors were 65% (273/422), 25% (6/24) and 75% (83/111) respectively. For outdoor resting *An. gambiae*, *An. arabiensis* and *An. funestus* the HBI was 46% (18/39), 25% (2/8) and 75% (3/4) respectively. In Kisian, the overall HBI of *An. gambiae*, *An. arabiensis* and *An. funestus* was 60, 7 and 83% respectively. The HBI for *An. gambiae* resting outdoors was 100% (3/3 and 60% (46/77) for indoor collections. The HBI for indoor and outdoor resting *An. arabiensis* was 7% (8/114) vs 6% (2/32). The HBI of *An. funestus* resting indoors was 83% (19/23).

## Sporozoites infection rates

A total of 1,517 samples comprising of 1,156 *An. gambiae* s.l and 361 *An. funestus* specimens were tested for *Plasmodium falciparum* Circumsporozoite (CSP) (Table 3). Ninety-one samples (90 Bungoma and 1 Kisian) tested positive giving an overall infection rate of 8% in Bungoma and 0.3% in Kisian. Overall, the sporozoite rate was higher indoors (9%, 82/956) than outdoors (4%, 8/190) in Bungoma, whereas in Kisian the sporozoite rate was 0.3%, 1/332) indoors. None of the samples collected outdoors in Kisian tested positive (n = 73). The sporozoite for *An. funestus* resting indoors was 11% (34/311). None of *An. funestus* collected outdoors was positive. The sporozoite rate for indoor resting *An. gambiae* was 8% (47/618) and outdoors at 5% (7/148). For *An. arabiensis* caught resting indoors, the sporozoite rate was 4% (1/27) and 3% (1/35) for outdoors. In Kisian, only 1/112 (1%) *An. gambiae* collected from indoor was CSP positive. No CSP positives were detected in *An. arabiensis* and *An. funestus* resting indoors and outdoors in Kisian. The overall entomological inoculation rates (EIRs) of

**Table 3. Sporozoite rates of *Anopheles* mosquitoes from indoor and outdoor collections in Bungoma and Kisian, western Kenya.**

| Site | Season | Location | *An. gambiae s.s* | | *An. arabiensis* | | *An. funestus* | |
|---|---|---|---|---|---|---|---|---|
| | | | N tested | Pf +ve | N tested | Pf +ve | N tested | Pf +ve |
| Bungoma | | Indoor | 618 | 47(8) | 27 | 1(4) | 311 | 34(11) |
| | | Outdoor | 148 | 7(5) | 35 | 1(3) | 7 | 0 |
| | Dry | | 290 | 21(7) | 26 | 0 | 100 | 18(18) |
| | Rainy | | 476 | 33(7) | 36 | 2(6) | 218 | 17(8) |
| Kisian | | Indoor | 112 | 1(1) | 147 | 0 | 73 | 0 |
| | | Outdoor | 16 | 0 | 54 | 0 | 3 | 0 |
| | Dry | | 41 | 1(2) | 127 | 0 | 47 | 0 |
| | Rainy | | 87 | 0 | 74 | 0 | 29 | 0 |

Pf, *Plasmodium falciparum*

+ve, Positive

the three vector species from indoor resting collections and outdoor in Bungoma was 66 and 10 infective bites/person/year respectively. In Kisian the overall EIR from indoor collections was 1.2 infective bites/person/year.

## Discussion

Given the widespread occurrence of pyrethroid insecticide resistance in malaria vectors in western Kenya [17,19,42], little is known about the behavioural response of these mosquito populations to the wide use of LLINs. Evidence has shown that successful malaria elimination strategies require vector control interventions that target the changing vector behaviour [43]. Overall, this study investigated the behavioral heterogeneity of malaria vectors for resting behavior, feeding choices and infection rates in the context of increased use of LLINs. The study revealed high resting densities, infection rates and insecticide resistance indoors compared to outdoors.

Of the three major malaria vectors in western Kenya, *An. gambiae* was the major vector in Bungoma followed by *An. funestus*. In Kisian *An. arabiensis* was predominant species followed by *An. gambiae*. The predominance of the two malaria vectors in the two different ecological settings is consistent with previous studies [16,44,45] observing increased frequency of *An. gambiae* s.s at sites further away from the lake Victoria basin and increase in the frequency of *An. arabiensis* at sites around the lake basin. There was a rise in the abundance of *An. funestus s.s* in Bungoma, a phenomena that has been observed previously in some regions in western Kenya [20]. Despite the high coverage and usage of long-lasting insecticidal nets (LLIN) in the region, increased indoor resting tendency of *An. gambiae* and *An. funestus* was observed, while *An. arabiensis* was found mostly resting outdoors. The variation in the relative frequency and behaviour of the three vectors has been observed earlier from the region [23,46]. It is worth noting that, the proportion of *An. arabiensis* was higher during the dry season in Kisian (lowland). This vector has been shown to survives best in areas with lower relative humidy and high temperatures and therefore prefers the lowland areas with such climates compared to the highland areas. The dry season also has low relative humidity and high temperatures [47].

This study observed reduced susceptibility of the F1 progeny of *An. funestus* and *An. gambiae* s.l captured resting indoors compared to those resting outdoors to deltamethrin insecticide. Earlier studies from western Kenya have reported increasing phenotypic resistance of *An. gambiae* s.l to pyrethroids [16,45,48,49]. Few studies have reported on *An. funestus* insecticide resistance[50] probably because of difficulties associated with rearing the larvae. The frequency

of Vgsc-1014S and Vgsc-1014F from indoors resting *An. gambiae* was higher than outdoors, from all sites. The Vgsc-1014F mutation was only detected in *An. gambiae* resting indoors, this mutation was absent in the mosquitoes collected outdoors. This might be that mosquitoes with some mutations are able to rest indoors in the presence of insecticides for vector control. Previous studies have observed the occurrence of Vgsc-1014F at low frequencies in East Africa including Kenya [42,51,52]. The presence of Vgsc-1014F in indoor resting mosquitoes may be of particular concern given that the mutation has been found to be strongly associated with pyrethroid resistance in West Africa [53]. The presence of kdr mutations at Bungoma where it was first detected in Kenya in *An. gambiae* [16] and now at the lowland site of Kisian indicate the widespread occurrence of the mutations among the *An. gambiae* population. The increased resistance level could be a result of selection pressure from insecticides used in vector control such as the widespread use of LLINs leading to the selection of resistant strains [6]. Some studies have shown a relationship between the spread of kdr alleles with the use of LLINs [54–56]. The extensive use of agriculture insecticides may also contribute to the occurrence of new mutations to existing insecticides [57–61].

Even though we observed high frequencies of Vgsc-1014S in *An. gambiae*, the allele was at low frequency in *An. arabiensis*, with a higher frequency of Vgsc-1014S, detected for indoor resting individuals than outdoors. Most recently, the presence of kdr mutations in *An. arabiensis* from the lowlands of western Kenya has been reported [17,49]. The low kdr frequency observed in *An. arabiensis* could be due to the reduced insecticide selection pressure as they resort to feed and rest outdoors in the absence of insecticides, unlike *An. gambiae* that feeds and rests indoors. The high frequency of kdr mutations, behavioural resilience and an increased proportion of *An. arabiensis* resting outdoors could all raise further concerns on the future utility of the current indoor interventions.

Sporozoite infection rates were high in *An. funestus* and *An. gambiae* collected from Bungoma, with *An. funestus* showing considerably higher sporozoite rates than the other species. These findings are in agreement with previous studies that observed high sporozoite rates in *An. funestus* and *An. gambaie*, implying the importance of the two vectors in malaria transmission in the region [20,46]. The higher sporozoite rates in *An. funestus* in this study further indicates its reemergence and as one of the primary important vector in malaria transmission in the region. The blood meal analysis showed a large proportion of the two species preferred feeding on humans than animals, with *An. funestus* observed to be highly anthropophagic in the region. This consistency in host choice has been frequently observed in *An. funestus* in Kenya and other parts of Africa [46,62,63]. This human-host choice and higher indoor resting proportions of *An. funestus* together with increased resistance poses a great concern in malaria elimination efforts due to its efficiency in transmitting malaria. These findings confirm earlier reports from other regions in western Kenya that have documented the re-emergence of *An. funestus* and its role in malaria transmission [20,64]. The infection rates were higher for vectors collected indoors than outdoors, suggesting ongoing malaria transmission regardless of the use of LLINs. Earlier reports from western Kenya have shown increased EIR for indoor collected mosquitoes [46]. Some studies have linked the rebound of malaria in western Kenya with increasing insecticide resistance after high coverage of LLINs [54,56,65]. The observed sporozoite infection rates outdoors might be attributed to changing in the biting behaviour of malaria vectors as some vectors could be feeding on humans when they are active and unprotected outdoors [66]. Complementary malaria control intervenions are thus, needed to control outdoor resting and biting mosquitoes, as the current tools only target indoor resting and biting mosquitoes.

Despite the low sporozoite rates of *An. arabiensis* reported in this study, its importance in outdoor malaria transmission should not be taken for granted due to its opportunistic

behavior, as the vector could continue with transmission outdoors in the region, which is a major threat to effective malaria vector control.

## Conclusion

The study shows high densities of *An. gambiae* and *An. funestus* resting indoors despite the use of indoor interventions with the increasing importance of *An. funestus* in sustaining malaria transmission in western Kenya highlands. *An. arabiensis* were more outdoors than indoors. This behavioural plasticity increases its survival and potential in continuing residual transmission after the main endophilic and endophagic vectors have been reduced by the interventions. The Vgsc-1014S and Vgsc-1014F mutations were observed at high frequencies in *An. gambiae* resting indoors. This calls for further screening of other resistance mutations in this population for better resistance management. Sporozoite rates were higher indoors than outdoors, showing that transmission occurs more indoors than outdoors in these sites. Insecticide resistance management strategies and/or new vector control interventions that may not be insecticide based are needed in western Kenya to reduce malaria transmission.

## Supporting information

**S1 Table. Summary of female *Anopheles* mosquitoes collected from indoor and outdoor in highland (Bungoma) and lowland (Kisian) settings of western Kenya.**
(DOCX)

**S2 Table. Summary results of KDR alleles for indoor and outdoor resting mosquitoes in Bungoma and Kisian sites in western Kenya.**
(DOCX)

## Acknowledgments

The authors wish to thank the villagers and community leaders in Bungoma and Kisian for their permission to collect mosquitoes in their houses. We acknowledge the Entomology Laboratory at Kenya Medical Research Institute, Kisumu for providing technical and laboratory space for the study. The permission to publish this study was granted by the director of Kenya Medical Research Institute.

## Author Contributions

**Conceptualization:** Maxwell G. Machani, Eric Ochomo, Guiyun Yan, Yaw A. Afrane.

**Data curation:** Maxwell G. Machani.

**Formal analysis:** Maxwell G. Machani, Jackline Kosgei, Guofa Zhou.

**Funding acquisition:** Guiyun Yan, Yaw A. Afrane.

**Investigation:** Maxwell G. Machani, Eric Ochomo.

**Methodology:** Maxwell G. Machani, Eric Ochomo, Yaw A. Afrane.

**Project administration:** Eric Ochomo, Yaw A. Afrane.

**Resources:** Guiyun Yan, Yaw A. Afrane.

**Software:** Guofa Zhou.

**Supervision:** Eric Ochomo, Fred Amimo, Stephen Munga, Guofa Zhou, Andrew K. Githeko, Guiyun Yan, Yaw A. Afrane.

**Validation:** Eric Ochomo, Guofa Zhou, Guiyun Yan, Yaw A. Afrane.

**Visualization:** Maxwell G. Machani, Eric Ochomo, Guofa Zhou, Yaw A. Afrane.

**Writing – original draft:** Maxwell G. Machani, Eric Ochomo, Fred Amimo, Jackline Kosgei, Stephen Munga, Andrew K. Githeko.

**Writing – review & editing:** Maxwell G. Machani, Eric Ochomo, Fred Amimo, Jackline Kosgei, Stephen Munga, Guofa Zhou, Andrew K. Githeko, Guiyun Yan, Yaw A. Afrane.

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
