## [Decision Letter · Decision Letter 0]

13 Nov 2019

PONE-D-19-29030

Resting behaviour of malaria vectors in a highland and a lowland site of western Kenya: Implication on malaria vector control measures

PLOS ONE

Dear Dr Afrane,

Thank you for submitting your manuscript to PLOS ONE. After careful consideration, we feel that it has merit but does not fully meet PLOS ONE’s publication criteria as it currently stands. Therefore, we invite you to submit a revised version of the manuscript that addresses the points raised during the review process.

The paper has now been seen by two reviewers. Both require some rewriting. Reviewer 1 suggests some extensive rewriting of the discussion but also some important rewording and structuring of the results. I agree with reviewer 1. Also please do not report values like '57.3' - only two digits are meaningful, so instead such a value should simply read 57. This in addition to the comments by the reviewers will make the paper an easier read.

We would appreciate receiving your revised manuscript by Dec 28 2019 11:59PM. To enhance the reproducibility of your results, we recommend that if applicable you deposit your laboratory protocols in protocols.io, where a protocol can be assigned its own identifier (DOI) such that it can be cited independently in the future. For instructions see: http://journals.plos.org/plosone/s/submission-guidelines#loc-laboratory-protocols

We look forward to receiving your revised manuscript.

Kind regards,

Friedrich Frischknecht

Academic Editor

PLOS ONE

Journal Requirements:

" This study was supported by grants from the National Institute of Health (R01 A1123074, U19 AI129326, R01 AI050243, D43 TW001505)

The funders had no role in study design, data collection and analysis, decision to publish, or preparation of the manuscript.".

i) Please provide an amended statement that declares *all* the funding or sources of support (whether external or internal to your organization) received during this study, as detailed online in our guide for authors at http://journals.plos.org/plosone/s/submit-now.  Please also include the statement “There was no additional external funding received for this study.” in your updated Funding Statement.

ii) Please include your amended Funding Statement within your cover letter. We will change the online submission form on your behalf.

3. We note that [Figure 1] in your submission contains [map/satellite] images which may be copyrighted. All PLOS content is published under the Creative Commons Attribution License (CC BY 4.0), which means that the manuscript, images, and Supporting Information files will be freely available online, and any third party is permitted to access, download, copy, distribute, and use these materials in any way, even commercially, with proper attribution. For these reasons, we cannot publish previously copyrighted maps or satellite images created using proprietary data, such as Google software (Google Maps, Street View, and Earth). For more information, see our copyright guidelines: http://journals.plos.org/plosone/s/licenses-and-copyright.

You may seek permission from the original copyright holder of Figure [1] to publish the content specifically under the CC BY 4.0 license. 

If you are unable to obtain permission from the original copyright holder to publish these figures under the CC BY 4.0 license or if the copyright holder’s requirements are incompatible with the CC BY 4.0 license, please either i) remove the figure or ii) supply a replacement figure that complies with the CC BY 4.0 license. Please check copyright information on all replacement figures and update the figure caption with source information. If applicable, please specify in the figure caption text when a figure is similar but not identical to the original image and is therefore for illustrative purposes only.

4.

We note that you have indicated that data from this study are available upon request. PLOS only allows data to be available upon request if there are legal or ethical restrictions on sharing data publicly. For more information on unacceptable data access restrictions, please see http://journals.plos.org/plosone/s/data-availability#loc-unacceptable-data-access-restrictions.

5. Please ensure that you refer to Figure 1 in your text as, if accepted, production will need this reference to link the reader to the figure.

6. Your ethics statement must appear in the Methods section of your manuscript. If your ethics statement is written in any section besides the Methods, please move it to the Methods section and delete it from any other section. Please also ensure that your ethics statement is included in your manuscript, as the ethics section of your online submission will not be published alongside your manuscript.

Reviewers' comments:

Reviewer's Responses to Questions

**Comments to the Author**

1. Is the manuscript technically sound, and do the data support the conclusions?

Reviewer #1: Partly

Reviewer #2: Yes

2. Has the statistical analysis been performed appropriately and rigorously? 

Reviewer #1: No

Reviewer #2: Yes

3. Have the authors made all data underlying the findings in their manuscript fully available?

Reviewer #1: No

Reviewer #2: Yes

4. Is the manuscript presented in an intelligible fashion and written in standard English?

Reviewer #1: Yes

Reviewer #2: Yes

5. Review Comments to the Author

Reviewer #1: The study by Afrane and colleagues investigates resting behavior, host preference and Pf infection of malaria vectors in two settings of western Kenya. The paper is in parts fairly well written; in other parts very repetitive and unstructured. The methodologies are routine in field entomology, the findings are robust and valuable. They support a large body of evidence of high levels of resistance in the area.

The paper would benefit from some serious re-writing but the paper The intro is gives detailed background information without being overly long. It is also well written and would not need much work. The Results, on the other hand, can be greatly shortened. A lot of details are provided in the long first paragraph. All percentages that are reported should come with some indicator of precision. Either a confidence interval around the proportion or mentioning numerator and denominator (n/N). The discussion section is not very focused and would benefit from re-writing. It could, without problems, be shortened with 25-33% to make the paper more attractive, focusing on time trends in resistance or feeding behavior in Western Kenya and the implications for control. The structure of the discussion can also be greatly improved: dealing with each of the important elements in a single clearly structured paragraph (instead of mixing all messages throughout the discussion) would really improve this part of an, otherwise valuable, paper.

SPECIFIC COMMENTS

The abstract lists many findings without a very clear selection or any statistical tests. It could be improved by presenting only the key results: sporozoite rate, tube assay (?), resistance genetic findings, and present them in a bit more detail and with stats.

Abstract. Trapping methods may be listed in the abstract.

Abstract: confidence intervals or details of the underlying number of mosquitoes (n/N) should be mentioned in the abstract. Also a statement ‘None of the outdoor collected mosquitoes in Kisian tested positive for sporozoite infections’ can only be interpreted if the number of examined mosquitoes is given.

Abstract: .’Vgsc-1014S was observed at a slightly higher frequency in An. gambiae s.s hereafter(An. gambiae) resting indoor than outdoor (89.7 vs 84.6% and 71.5 vs 61.1%) in Bungoma and Kisian respectively’ is unclear. Vgsc is not defined in the abstract. And the estimates are possibly not statistically significant. If so, the finding should receive less prominence in the results section.

Abstract: ‘high densities of insecticide-resistant… [mosquitoes]’ in the conclusion can only be presented if also mosquito burden is presented in the abstract. What was the average number of mosquitoes per trapping night? It is really high? Or did the authors (simply) want to say that among mosquitoes there was a large proportion of resistant mosquitoes?

Intro, line 112. ‘Insecticide resistance could make mosquitoes respond numerically…’ is unclear to me. What is intended? This sentence could be simplified, probably by splitting the sentence in two.

Methods, line 152. Clay pots are notoriously difficult to standardize. A useful reference is Van den Bijllaardt, Acta Tropica 111 (2009) 197–199 where very detailed procedures are explained. For the current method, more details should be given on where traps were placed and how sampling was done. This applies to all trapping methods, not just clay pots.

Methods, line 195/196. The authors may want to comment on reports from SE Asia (Coosemans and team) that there is cross-reactivity in ELISA results in mosquitoes that fed on cattle blood. This might have contributed to the fairly high sporozoite rates observed in this study.

Methods: the dry versus wet season sampling is useful but needs some indication to where in the season samples were taken. How many weeks after start of the rains was ‘wet season’? How dry was the dry season?

Results, line 218-220. ‘Overall, the proportion of Anopheles species resting indoors was significantly higher by 82.4% than outdoor location 17.6% across the study sites (z = -8.47, p < 0.0001) ‘. This is very imprecise. I am not sure what is compared here. Is it the proportion of anophelines among caught mosquitoes? Or is it the absolute number of anophelines caught indoors vs outdoors? If so: is this appropriate? The sampling surface was probably very different. Only sporozoite rates and blood meal sources can be compared indoors vs outdoors since trapping methods differed profoundly.

Similarly, line 232 is unclear ‘There were more An. gambiae and An. funestus resting indoors than outdoors (80.7 vs 19.3% and 97.8 vs 2.2% respectively; X2 =122.96, df = 2, p < 234 0.0001). ‘ How can this be a percentage? Is it the percentage of all gambiae that was caught indoors? I find these presentations very difficult and dangerous since the sampling intensity was so different. I would advise to simply present for indoor and outdoor sampled mosquitoes separately: numbers per trapping night, species composition, sporozoite rates, resistance phenotype, genotype, blood source. Compare the % sporozoite positive between indoor vs outdoor but do not compare absolute numbers! Reconsidering what the authors really want to present would also allow this very long and difficult first section of the results section to be shortened considerably. That would really improve the paper. At the moment, it is the weaker part of the paper. Make use of the display figures (tables and figures) and do not add too much detail on proportions etc in the main text.

Throughout the results, give n/N or confidence intervals. Percentages without any indication of precision (either numbers of confidence intervals) are not very informative.

For blood sources, the authors make statements on higher vs lower between species and indoor vs outdoor. Present statistics and indicate precision of estimates. If a finding is not statistically significant, do not claim differences.

I am puzzled by Figure 2. There seems to be much less variation in mosquito exposure than is typically reported. Are these only successful trapping efforts or are zeroes (no mosquitoes caught) also included in the figure? These findings typically follow a Poisson or Negative Binomial distribution. The current presentation suggests a normal distribution with very few negative trapping events. That would be unusual. It is important to explain this and present the % of traps with at least one mosquito and the distribution of mosquitoes in positive traps.

Figure 3: adjust title. This figure doesn’t present abundance. Only composition.

Discussion

Line 331 ‘…leading to increased vector-human contact and ongoing malaria transmission in the region despite the high coverage and usage of LLINs’ is not an appropriate sentence. This was in no way proven by the data. It is pure speculation and should be recognizable as such (and probably placed somewhere later in the discussion and not in the crucial first paragraph).

As indicated above, the discussion is slightly unstructured. Findings and interpretations are mixed. It would be recommended to really work in clearly structured paragraphs that could explain in detail and with good references:

Paragraph 1. Conclusion. Pretty much as it is now.

Paragraph 2. Species composition in the areas. How does this relate to other findings?

Paragraph 3. How does resistance differ between species?

Paragraph 4. The resistance phenotype and genotype indoors and outdoors. Did this change compared to earlier studies in the area? Is this an indicator that biting increases indoors as a consequence of increased resistance (which again could be because of increased pressure due to widescale use of interventions)

Paragraph 5. Sporozoite rate per species among outdoor mosquitoes and for wet versus dry season. How does this compare to other studies? What does it mean for what vector is most important for transmission in the region? Include the blood meal preference in this section. Its relevance lies in this (explaining human biting and thus vector importance)

Paragraph 6. Final conclusion.

Reviewer #2: This manuscript describes the composition, bionomics and epidemiology of Anopheles malaria vectors in Western Kenya. The work is technically well done. The results are valuable and interesting, because malaria transmission continues and has even rebounded in parts of Africa despite decades of vector control.

The manuscript is well written and contains thoughtful analysis. It can be published after only several very minor points that should be corrected, as follows.

line 199 (and other locations): "anopheline" is an adjective, not a Latin name, and should not be italicized or capitalized.

line 252: unclear wording where it says "High resistance levels" of 36.6% or 65.5%, but the numbers refer to the 24 hour mortality rate, not actually resistance. Please reword this.

line 319, 320 (and other locations): be consistent about Latin names. Names should be spelled out at first use, and thereafter genus is abbreviated.

line 352 (and other locations): be consistent about "kdr", which is not capitalized, but here is shown as Kdr

6. PLOS authors have the option to publish the peer review history of their article (what does this mean?). If published, this will include your full peer review and any attached files.

Reviewer #1: No

Reviewer #2: No

---

## [Author Response · Author response to Decision Letter 0]

22 Jan 2020

``Response to reviewer comments

The following is our response to the comments by the reviewers. We really thank reviewers for their constructive criticism and insights that has helped to improve this paper.

Reviewer #1: 

The study by Afrane and colleagues investigates resting behavior, host preference and Pf infection of malaria vectors in two settings of western Kenya. The paper is in parts fairly well written; in other parts very repetitive and unstructured. The methodologies are routine in field entomology, the findings are robust and valuable. They support a large body of evidence of high levels of resistance in the area.

The paper would benefit from some serious re-writing but the paper The intro is gives detailed background information without being overly long. It is also well written and would not need much work. The Results, on the other hand, can be greatly shortened. A lot of details are provided in the long first paragraph. All percentages that are reported should come with some indicator of precision. Either a confidence interval around the proportion or mentioning numerator and denominator (n/N). The discussion section is not very focused and would benefit from re-writing. It could, without problems, be shortened with 25-33% to make the paper more attractive, focusing on time trends in resistance or feeding behavior in Western Kenya and the implications for control. The structure of the discussion can also be greatly improved: dealing with each of the important elements in a single clearly structured paragraph (instead of mixing all messages throughout the discussion) would really improve this part of an, otherwise valuable, paper.

Specific comments

Abstract

Reviewer: The abstract lists many findings without a very clear selection or any statistical tests. It could be improved by presenting only the key results: sporozoite rate, tube assay (?), resistance genetic findings, and present them in a bit more detail and with stats.

Response: The result section has been revised and only key findings i.e. [Species composition per study site, indoor and outdoor resting densities with statistical evidence, phenotypic resistance (WHO tube assays) with confidence intervals, genotypic resistance per site per species and sporozoite infections] have been presented in details

Reviewer: Trapping methods may be listed in the abstract

Response: The trapping methods have been included in the methods section in Line 41-43

Reviewer: Confidence intervals or details of the underlying number of mosquitoes (n/N) should be mentioned in the abstract. Also a statement ‘None of the outdoor collected mosquitoes in Kisian tested positive for sporozoite infections’ can only be interpreted if the number of examined mosquitoes is given.

Response: The confidence intervals around the percentages, numerator and denominator have been included in the result section. Lines 51-67. Number of outdoor resting mosquitoes analysed for sporozoite infection in Kisian has been included in Line 69

Reviewer: Vgsc-1014S was observed at a slightly higher frequency in An. gambiae s.s hereafter(An. gambiae) resting indoor than outdoor (89.7 vs 84.6% and 71.5 vs 61.1%) in Bungoma and Kisian respectively’ is unclear. Vgsc is not defined in the abstract. And the estimates are possibly not statistically significant. If so, the finding should receive less prominence in the results section.

Response: This finding has been deleted from the abstract.

Reviewer: high densities of insecticide-resistant… [mosquitoes]’ in the conclusion can only be presented if also mosquito burden is presented in the abstract. What was the average number of mosquitoes per trapping night? It is really high? Or did the authors (simply) want to say that among mosquitoes there was a large proportion of resistant mosquitoes?

Response: This statement has been revised and now it reads “….high indoor resting densities of An. gambiae and An. funestus, insecticide resistance, and persistence of malaria transmission indoors with high entomological inoculation rates (EIR) regardless of the use of Long-lasting insecticidal nets (LLINs)”.

Introduction

Reviewer: line 112. ‘Insecticide resistance could make mosquitoes respond numerically…’ is unclear to me. What is intended? This sentence could be simplified, probably by splitting the sentence in two.

Response: The sentence has been deleted from the introduction

Methods

Reviewer: line 152. Clay pots are notoriously difficult to standardize. A useful reference is Van den Bijllaardt, Acta Tropica 111 (2009) 197–199 where very detailed procedures are explained. For the current method, more details should be given on where traps were placed and how sampling was done. This applies to all trapping methods, not just clay pots

Response: This has been revised with more details given in the methodology section for all trapping methods. Van den Bijllaardt et al 2009 and Odiere et al 2007 have been added as the most relevant reference for clay pot collections. Line 152-166

Reviewer: line 195/196. The authors may want to comment on reports from SE Asia (Coosemans and team) that there is cross-reactivity in ELISA results in mosquitoes that fed on cattle blood. This might have contributed to the fairly high sporozoite rates observed in this study.

Response: It is true our study observed high sporozoite rates, this may not be as a result of cross reactivity in ELISA assays with cattle blood as the majority of the samples were fed on human blood in Bungoma. If there was cross-reactivity we could have experienced it in Kisian where a large number of collected mosquitoes were zoophilic and have taken cattle blood with low sporozoite rates observed.

Reviewer: The dry versus wet season sampling is useful but needs some indication to where in the season samples were taken. How many weeks after start of the rains was ‘wet season’? How dry was the dry season?

Response: The details on the dry and wet season has been included in the methodology section. “…

“Mosquitoes were sampled during the middle of long dry season (February-March) and four weeks after the start of long rainy season (May-July)” Meaning mosquitoes were collected after 4 weeks the start of the hot dry and long rain season. Line 150-151

Reviewer: line 218-220. ‘Overall, the proportion of Anopheles species resting indoors was significantly higher by 82.4% than outdoor location 17.6% across the study sites (z = -8.47, p < 0.0001) ‘. This is very imprecise. I am not sure what is compared here. Is it the proportion of anophelines among caught mosquitoes? Or is it the absolute number of anophelines caught indoors vs outdoors? If so: is this appropriate? The sampling surface was probably very different. Only sporozoite rates and blood meal sources can be compared indoors vs outdoors since trapping methods differed profoundly.

Response: That sentence on the comparison has been deleted from the result section.

Reviewer: Similarly, line 232 is unclear ‘There were more An. gambiae and An. funestus resting indoors than outdoors (80.7 vs 19.3% and 97.8 vs 2.2% respectively; X2=122.96, df = 2, p < 234 0.0001). ‘ How can this be a percentage? Is it the percentage of all gambiae that was caught indoors? I find these presentations very difficult and dangerous since the sampling intensity was so different. I would advise to simply present for indoor and outdoor sampled mosquitoes separately: numbers per trapping night, species composition, sporozoite rates, resistance phenotype, genotype, blood source. 

Response: The section now presents the resting densities of malaria vectors indoors and outdoors, species composition per study site, overall malaria vector composition per location and seasonal species composition.

Reviewer: Compare the % sporozoite positive between indoor vs outdoor but do not compare absolute numbers! Reconsidering what the authors really want to present would also allow this very long and difficult first section of the results section to be shortened considerably. That would really improve the paper. At the moment, it is the weaker part of the paper. Make use of the display figures (tables and figures) and do not add too much detail on proportions etc in the main text.

Response: This result section has been revised according to reviewers suggestions. The n/N for sporozoite percentages is included in the table. Table 3

Reviewer: Throughout the results, give n/N or confidence intervals. Percentages without any indication of precision (either numbers of confidence intervals) are not very informative.

Response: The confidence intervals around the percentages, n/N have been included in the text and also in the tables

Reviewer: For blood sources, the authors make statements on higher vs lower between species and indoor vs outdoor.present statistics and indicate precision of estimates. If a finding is not statistically significant, do not claim differences

Response: The statements indicating higher and lower in the blood meal findings has been revised throughout the text and the precision of estimates (n/N) included in the table. Table 2

Reviewer: I am puzzled by Figure 2. There seems to be much less variation in mosquito exposure than is typically reported. Are these only successful trapping efforts or are zeroes (no mosquitoes caught) also included in the figure? These findings typically follow a Poisson or Negative Binomial distribution. The current presentation suggests a normal distribution with very few negative trapping events. That would be unusual. It is important to explain this and present the % of traps with at least one mosquito and the distribution of mosquitoes in positive traps.

Response: The figure 2 has been changed and it now presents successful trapping efforts.

Reviewer: Figure 3: adjust title. This figure doesn’t present abundance. Only composition.

Response: This has been revised and now reads “ Seasonal and Anopheles sibling species composition resting indoors and outdoors in A: Highland site (Bungoma) and B: lowland site (Kisian), western Kenya”

Discussion

Reviewer: Line 331 ‘…leading to increased vector-human contact and ongoing malaria transmission in the region despite the high coverage and usage of LLINs’ is not an appropriate sentence. This was in no way proven by the data. It is pure speculation and should be recognizable as such (and probably placed somewhere later in the discussion and not in the crucial first paragraph).

Response: The sentence has been revised to read “The study revealed high resting densities, infection rates and insecticide resistance indoors compared to outdoors”

Reviewer: As indicated above, the discussion is slightly unstructured. Findings and interpretations are mixed. It would be recommended to really work in clearly structured paragraphs that could explain in detail and with good references:

Paragraph 1. Conclusion. Pretty much as it is now.

Paragraph 2. Species composition in the areas. How does this relate to other findings?

Paragraph 3. How does resistance differ between species?

Paragraph 4. The resistance phenotype and genotype indoors and outdoors. Did this change compared to earlier studies in the area? Is this an indicator that biting increases indoors as a consequence of increased resistance (which again could be because of increased pressure due to widescale use of interventions)

Paragraph 5. Sporozoite rate per species among outdoor mosquitoes and for wet versus dry season. How does this compare to other studies? What does it mean for what vector is most important for transmission in the region? Include the blood meal preference in this section. Its relevance lies in this (explaining

human biting and thus vector importance)

Paragraph 6. Final conclusion.

Response: This has been addressed and discussed further according to the reviewer’s suggestion . Paragraph 1 Presents the significance of the study and the main findings of the study.

Paragragh 2: discussion on the species composition in the region and how it relates with previous work from the same region

Paragraph 3&4 discussion on the phenotypic and genotypic resistance per species per location(indoor and outdoor location) per study sites comparing with earlier findings

Paragragh 5 discussion on sporozoite rates per species among indoor and outdoor collections blood meal sources compared with earlier reports. The primary vector in the region.

Paragraph 6 final conclusion

Response to Reviewer # 2

This manuscript describes the composition, bionomics and epidemiology of Anopheles malaria vectors in Western Kenya. The work is technically well done. The results are valuable and interesting, because malaria transmission continues and has even rebounded in parts of Africa despite decades of vector control.

The manuscript is well written and contains thoughtful analysis. It can be published after only several very minor points that should be corrected, as follows.

Reviewer: Line 199 (and other locations): "anopheline" is an adjective, not a Latin name, and should not be italicized or capitalized.

Response: This has been revised throughout the text

Reviewer: line 252: unclear wording where it says "High resistance levels" of 36.6% or 65.5%, but the numbers refer to the 24 hour mortality rate, not actually resistance. Please reword this.

Response: Revision has been done and the sentence now reads, “Reduced mortality was observed for F1 progeny of Anopheles gambiae s.l resting indoors (37%, 95%......”

Reviewer: line 319, 320 (and other locations): be consistent about Latin names. Names should be spelled out at first use, and thereafter genus is abbreviated.

Response: Corrected, consistency checked throughout the text

Reviewer: Line 352 (and other locations): be consistent about "kdr", which is not capitalized, but here is shown as Kdr

Response: Corrected, consistency checked throughout the text “kdr” with lower case used

Technical Comments from the Editorial Office

Editor: Please do not report values like '57.3' - only two digits are meaningful, so instead such a value should simply read 57.

Response: This has been revised and the two digits value maintained throughout the result section. 

Editor: Please provide an amended statement that declares *all* the funding or sources of support (whether external or internal to your organization) received during this study, as detailed online in our guide for authors at http://journals.plos.org/plosone/s/submit-now

Response: There was no additional external funding received for this study.

Editor: We note that [Figure 1] in your submission contains [map/satellite] images which may be copyrighted. All PLOS content is published under the Creative Commons Attribution License (CC BY 4.0), which means that the manuscript, images, and Supporting Information files will be freely available online, and any third party is permitted to access, download, copy, distribute, and use these materials in any way, even commercially, with proper attribution. For these reasons, we cannot publish previously copyrighted maps or satellite images created using proprietary data, such as Google software (Google Maps, Street View, and Earth). For more information, see our copyright guidelines: http://journals.plos.org/plosone/s/licenses-and-copyright.

Response: Figure 1 has been removed

Editor: We note that you have indicated that data from this study are available upon request. PLOS only allows data to be available upon request if there are legal or ethical restrictions on sharing data publicly. For more information on unacceptable data access restrictions, please see http://journals.plos.org/plosone/s/dataavailability# loc-unacceptable-data-access-restrictions.

Response: The statement on data availability has been included and now it reads…… “The datasets used for the current study are available at the repository of the Kenya Medical Research Institute”

Editor: Your ethics statement must appear in the Methods section of your manuscript. If your ethics statement is written in any section besides the Methods, please move it to the Methods section and delete it from any other section. Please also ensure that your ethics statement is included in your manuscript, as the ethics section of your online submission will not be published alongside your manuscript.

Response: The ethical statement has been moved to the methods section.

Editor: Please ensure that you refer to Table 2 in your text as, if accepted, production will need this reference to link the reader to the Table.

Response: Table 2 has been referred in the text

---

## [Editor Report · Decision Letter 1]

5 Feb 2020

Resting behaviour of malaria vectors in  highland and  lowland sites of western Kenya: Implication on malaria vector control measures

PONE-D-19-29030R1

Dear Dr. Afrane,

We are pleased to inform you that your manuscript has been judged scientifically suitable for publication and will be formally accepted for publication once it complies with all outstanding technical requirements.

With kind regards,

Friedrich Frischknecht

Academic Editor

PLOS ONE

Additional Editor Comments (optional):

Thanks for returning a well done revision, in my view you addressed all the important issues raised by the reviewers as well as their minor comments. After careful reading your revised manuscript, I thus decided not to send it to the reviewers again. There are still a few minor improvements possible in terms of wording (e.g. first sentence of abstract needs an 'and'; first sentence of discussion needs attention) and the reference list needs attention as the cited literature is not all in the journal style. I trust you can make this changes without a need for further external review. Congratulation to a nice study.
---

## [Editor Report · Acceptance letter]

11 Feb 2020

PONE-D-19-29030R1 

Resting behaviour of malaria vectors in highland and lowland sites of western Kenya: Implication on malaria vector control measures 

Dear Dr. Afrane:

I am pleased to inform you that your manuscript has been deemed suitable for publication in PLOS ONE. Congratulations! Your manuscript is now with our production department. 

With kind regards,

on behalf of

Dr. Friedrich Frischknecht 

Academic Editor

PLOS ONE